# Protected Areas and Rural Depopulation in Spain: A Multi-Stakeholder Perceptual Study

**David Rodríguez-Rodríguez** [1,*]  **and Remedios Larrubia Vargas** [2]

1. European Topic Centre, University of Malaga, 29010 Malaga, Spain
2. Department of Geography, University of Malaga, 29010 Malaga, Spain; rlarrubia@uma.es
* Correspondence: davidrr@uma.es; Tel.: +34-951953102

**Abstract:** Protected areas (PAs) are thought by some to contribute to local wellbeing and socioeconomic development, whereas for others PAs remain a regulatory burden that hampers rural development. Here, we sought to ascertain the perceived causes of rural depopulation and the potential impact of four Natura 2000 sites on the wellbeing and depopulation figures of four protected rural municipalities in Spain that were selected as extreme case studies. We used phone surveys to elicit experts' views ($n = 19$) on the topic and convened eight in-person workshops to garner local residents' insights ($n = 40$) using structured questionnaires. We complemented perceived wellbeing data from PAs with surveys to residents in neighbouring unprotected municipalities ($n = 28$). Both experts and workshops' attendees from protected municipalities overwhelmingly attributed depopulation figures to structural causes linked to transport accessibility, basic service provision and the existence of job opportunities, which they perceived to be unrelated to the PAs' regulations or management. Local residents did generally not perceive any impact on their collective or individual wellbeing from those PAs, and most who did, expressed a negative impact chiefly due to socioeconomic restrictions. Four-fifths of the experts and half of the workshops' attendees from protected municipalities, however, expressed that PAs' administrations could help improve depopulation figures in their towns mainly through promoting tourism and greater compatibility of land uses, including housing and infrastructure development. While the assessed Natura 2000 sites certainly have scope for tourism promotion, their lenient legal regimes make it largely unfeasible to broaden land use compatibility without damaging protected features.

**Keywords:** Natura 2000; impact; resident; expert; opinion; wellbeing; case study



## 1. Introduction

The loss of rural populations entails a number of important social, economic and environmental changes for territorial sustainability. Among them, the cessation of traditional economic activities linked to the primary sector, ageing, the reduced provision of basic services, ecological succession leading to landscape changes, increased wildfire risk, the loss of cultural practices, or shifts in the provision of ecosystem services are all significant potential changes [1–4]. Rural depopulation has been affecting communities in developed countries for a long time [4–6]. More recently, it is affecting developing countries as well [1]. Rural depopulation has been a long-lasting issue in Spain. Massive rural emigration to cities and other European countries in pursuit of better working and living conditions took place in the country in the 1950s and 1960s. Such trends decreased in the following decades, but never stopped or were reversed [7,8]. After Spanish accession to the European Union in 1986, a number of policies whose main or ancillary objectives were to enhance the wellbeing of rural communities and stop rural depopulation have been implemented with European funds, notably through the European Agricultural Fund for Rural Development of the Common Agricultural Policy [9], although its implementation has been uneven in Spain [7,10,11].

Other policies of broad impact on the territory relate to environmental conservation. Protected areas (PAs) cover over 17.46% of the land territory of the European Union [12] and more than one-third of the Spanish land territory [13], and are expected to increase globally to cover at least 30% of land and sea areas by 2030 [14]. PAs may provide opportunities for rural development linked to enhanced ecosystem services and tourism [15]. PA regulatory and managerial regimes may, in turn, entail restrictions to some human uses of the territory that may diminish local wellbeing, exacerbate poverty and increase depopulation of rural areas [16]. A previous country-wide study found that protected rural municipalities in Spain had had generally worse depopulation trends than neighbouring unprotected municipalities, although exceptions occurred, mostly related to municipalities in Sites of Community Importance (SCIs), which tended to perform better than their controls [17]. Although the authors carefully controlled for a number of bio-physical covariates that might influence rural depopulation, they could not accurately attribute the depopulation effects to PA regulations given that other unconsidered factors might have influenced the fact that some municipalities performed better than others at maintaining or increasing their populations.

Stakeholders' perceptions on conservation initiatives, notably those of the people affected by environmental policies and regulations, allow researchers to delve into the causes and consequences of such initiatives, assess their social acceptance and estimate their success [18,19]. In this study, we used stakeholders' perceptions to refine the findings of the previous country-wide study [17] by: (1) Ascertaining whether PA regulations and/or management may have influenced depopulation in a sample of pairs of protected and unprotected small rural municipalities with the most contrasting depopulation figures in Spain; (2) Determining if other factors may have influenced depopulation trends in those municipalities; (3) Making recommendations to rural development authorities and PA authorities for enhancing the wellbeing of rural communities and reducing or preventing depopulation.

## 2. Materials and Methods

### 2.1. Research Questions

In this study we sought to delve into the causes and possible solutions to the depopulation issue in rural Spain through three research questions:

1. Have the regulations or managerial regimes of PAs affected local wellbeing and rural depopulation in the selected municipalities?
2. Are there other factors that may have influenced rural depopulation in the selected municipalities?
3. What can be done to improve depopulation trends in the selected municipalities?

### 2.2. Studied Municipalities

Four pairs of extreme cases regarding the values of three depopulation indicators from the initial census sample of 52 protected municipalities and 55 unprotected municipalities of the study by [17] were chosen. The original census sample of 107 rural municipalities from which those cases were taken had defined a 'rural municipality' as those municipalities that had less than 10,000 inhabitants [7] and were located further from 20km from cities of 10,000 inhabitants or more [20]. The 52 protected municipalities had over 99% of their territories inside a multiple-use PA, thus being affected by their regulations and management, whereas unprotected municipalities had less than 1% of their territory affected by PA regulations and management. Of our sample of four extreme pairs of municipalities in terms of contrasting depopulation figures, three pairs related to SCIs and one pair related to a Special Protection Area (SPA), which were first designated on each site. They had been designated between March of 1999 and January of 2001. Yearly municipal population data from 1996 until 2019 were retrieved from official sources [21], and three depopulation indicators were compared before and after the designation dates of each PA in case and control municipalities: Compound annual growth rate, proportion of reproductive individuals,

and proportion of reproductive females (Table 1). Data were retrieved until 2019 due to the unusual data from 2020 due to the effects of the COVID-19 pandemic.

**Table 1.** Selected pairs of Spanish rural municipalities with the most contrasting depopulation figures before and after each protected area was designated in the 1996 – 2019 period. CAGR: Compound Annual Growth Rate; PRI: Proportion of reproductive individuals; PRF: Proportion of reproductive females. SCI: Site of Community Importance; SAC: Special Areas of Conservation; SPA: Special Protection Area.

| Protected Area (Designation Date) | Protected Municipality | | | Unprotected Municipality | | |
|---|---|---|---|---|---|---|
| | CAGR (%) | PRI (%) | PRF (%) | CAGR (%) | PRI (%) | PRF (%) |
| Hoces del Cabriel, Guadazaon y ojos de Moya (2001-SCI; 2016-SAC) | La Pesquera | | | Puebla del Salvador | | |
| | −1.43 | −7.30 | −17.66 | 0.22 | 2.02 | 5.14 |
| Sabinares de Somosierra (2000-SCI; 2015-SAC) | Casla | | | Santa Marta del Cerro | | |
| | 0.43 | 12.44 | 17.93 | −2.26 | −16.73 | −10.05 |
| Sierras de Urbión y Cebollera (1999-SCI; 2015-SAC) | La Poveda de Soria | | | Vizmanos | | |
| | 1.97 | 1.67 | 16.11 | −2.44 | −13.86 | −18.36 |
| Penyagolosa (2000-SPA) | Chodos | | | Atzeneta del Maestrat | | |
| | −1.77 | −12.76 | −27.69 | 0.08 | −1.35 | −2.90 |

The four pairs of municipalities were located in four provinces belonging to three different Spanish regions: Castilla Leon, Castilla La Mancha and Valencia (Figure 1).

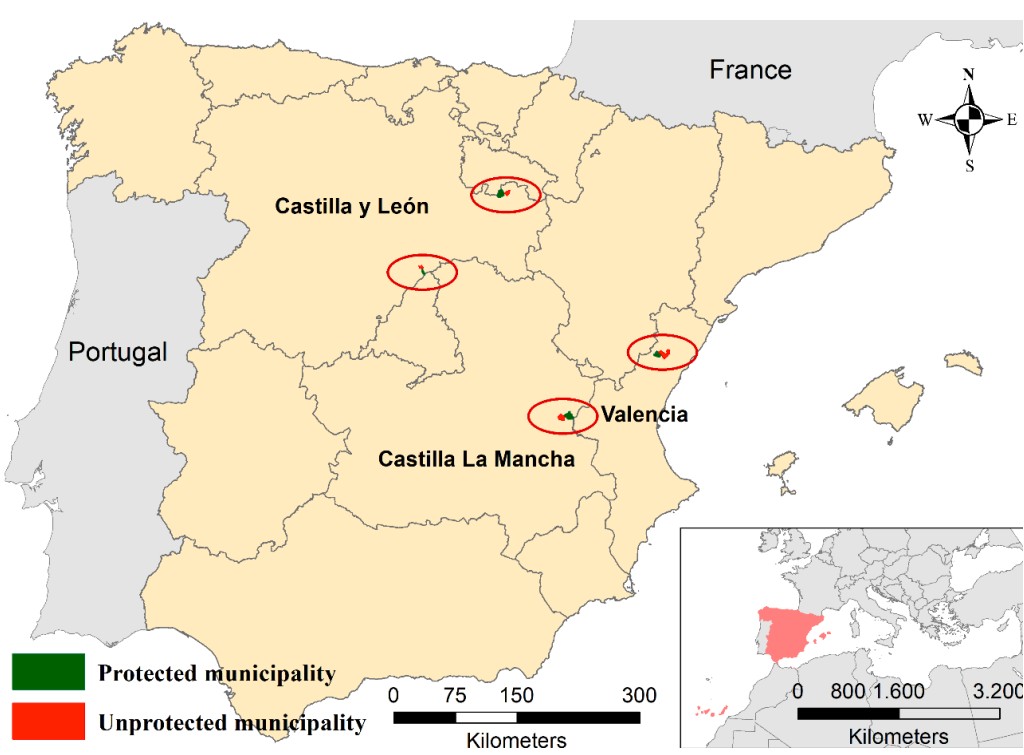

**Figure 1.** Location of the four pairs of municipalities in Spain.

*2.3. Data Gathering*

Two social groups, experts and local residents, were surveyed in protected municipalities in order to answer the research questions. Experts included town mayors, rural development officers from the provincial and regional administrations (provincial delegations), PA managers, and coordinators of rural development action groups. Rural development officers of the Valencia region could not be reached by phone despite numerous attempts. Local residents included 18 year-old and older people living in each of the protected and unprotected municipalities for most of the year or visiting them regularly outside main holiday periods for a number of years. Data from residents in unprotected municipalities were used to complement the results on the perceived impact of PAs on local wellbeing.

Two social science data compilation methods were used: (1) Experts were interviewed via telephone using a structured questionnaire (Annex S1); and (2) One workshop was held in each of the selected municipalities to garner the opinions of local residents using a similar structured questionnaire (Annex S2). A briefing explaining the background and objectives of the study was provided prior to each phone interview or workshop for context. Purposive samples of local residents were gathered in local councils' premises with the help of town mayors. COVID-19 prevention measures such as face masks and social distancing were taken at every workshop. A number of measures were taken to increase representation and participation of local residents. Firstly, guidance to increase representation of attendees to the workshops was sent to the mayors in advance via email (Annex S3). Based on such guidance, mayors were asked to identify potential attendees and invite them to attend the workshops. Moreover, a poster announcing the workshop was created for each of the municipalities and sent to mayors for greater dissemination of the event across the towns and broadened participation (Annex S4). All the workshops took place between 25 October and 4 November 2021.

## 3. Results

*3.1. Sample Data*

Nineteen experts responded to our phone interviews on the four protected rural municipalities (Table 2).

**Table 2.** Type of expert interviewed by phone on the protected municipalities.

| Expert Type | Number |
|---|---|
| Protected area manager | 4 |
| Town mayor | 4 |
| County rural development coordinator | 4 |
| Provincial Government—Rural development officer | 4 |
| Regional Government—Rural development officer | 3 |

Forty local inhabitants attended the workshops and filled in the questionnaires in the four protected municipalities. Fifty per cent of them were men and fifty per cent were women, and they ranged from 20 to 77 years old (mean age of 50.15 years). They had lived 28.8 years in town on average. Over half of them worked in the tertiary sector or were retired (Figure 2). Twenty-eight inhabitants from the four unprotected municipalities responded to the questionnaire. Fifty-seven per cent of them were men and 43% were women. Four questionnaires were rejected for having most fields or key fields incomplete or having been filled in by a neighbour from a different town.

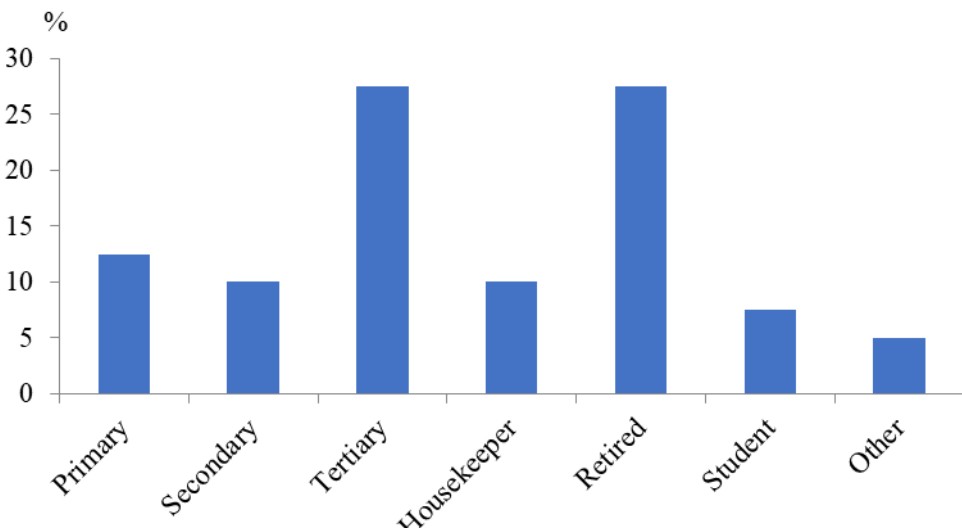

**Figure 2.** Occupations of respondents to the questionnaire in protected municipalities by sector, in percentage (*n* = 40).

### 3.2. Effects of PAs on Rural Depopulation

3.2.1. Experts

A total of 68.4% of the experts perceived little or no effect from PA regulations on depopulation of rural communities in PAs. A total of 21% of them perceived PAs to have had a positive impact against depopulation due to enhanced tourism activities and incoming subsidies, whereas 5% perceived negative impacts from PA designation. Half of the surveyed PA managers expressed that their Natura 2000 sites (one SCI and one SPA) had had no impact on depopulation, whereas the other two perceived negative or slightly negative impacts resulting from restrictions to economic activities.

A total of 68.4% of experts perceived no effect of PA management on rural depopulation, and 31.6% could not answer that question. All PA managers stated that PA management had not affected rural depopulation.

3.2.2. Local Populations

Ninety per cent of the workshops' attendees in the four protected municipalities stated that they perceived a trend towards depopulation in their municipalities prior to the COVID-19 pandemic in 2020. However, 70% of them stated that the PAs had had no impact on the residents' wellbeing. For 20% of the attendees, the PAs had had a negative impact on the towns' residents' wellbeing chiefly due to socioeconomic restrictions, whereas 7.5% expressed they had had a positive impact. A total of 80% of the attendees in protected municipalities perceived no impact of PAs on their own wellbeing, 15% stated a negative impact, and 5% expressed a positive impact.

Of the eight respondents perceiving negative collective impacts from PAs, three worked in the primary sector (farming or forestry), and two were retired. However, only half of them, including a pharmacy worker and a retired person, stated a negative impact of the PAs on their own wellbeing due to socioeconomic restrictions and complex administrative procedures. Of the three attendees stating positive collective impacts of PAs on the grounds of enjoying nature and rural conservation, one worked in the industry, one was disabled and the other was a cattle farmer. The farmer, however, stated negative personal impact from PAs due to restrictions to her activity.

Almost 93% of the workshops' attendees in unprotected municipalities expressed that the PAs' regulations had had no impact on the residents' wellbeing or on their own wellbeing. For 7.1% of them, PAs had had positive collective and individual impacts. One of the reasons for this was that PA regulations allowed for rubbish tips to deposit dead cattle.

### 3.3. Other Causes of Rural Depopulation

3.3.1. Experts

The main stated cause for the relatively good depopulation values in the two best performing protected municipalities was good transport infrastructure (Figure 3).

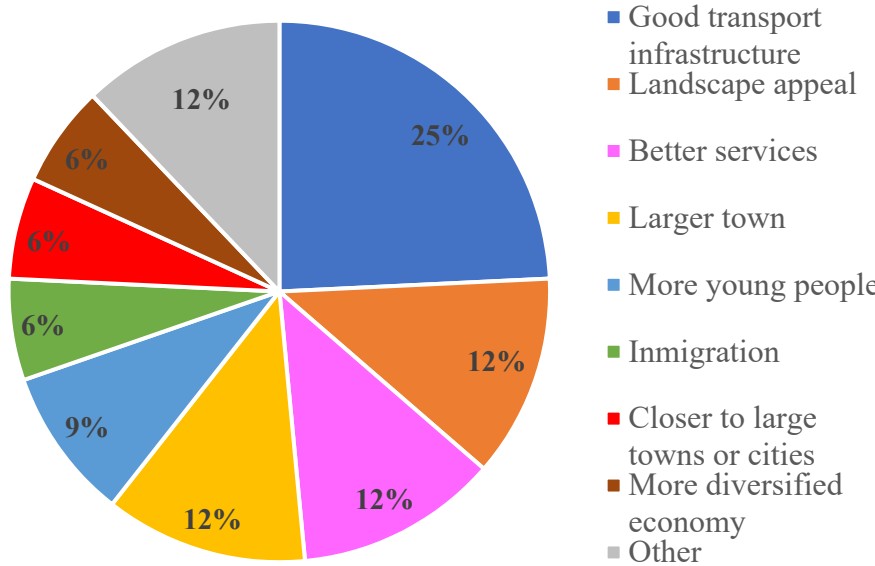

**Figure 3.** Stated causes for the depopulation figures in the two best performing protected municipalities, according to experts.

In contrast, a range of individual causes, including poor transport infrastructure, insufficient job opportunities and isolation were chiefly mentioned to explain the poorer depopulation values in the two worst performing protected municipalities (Figure 4).

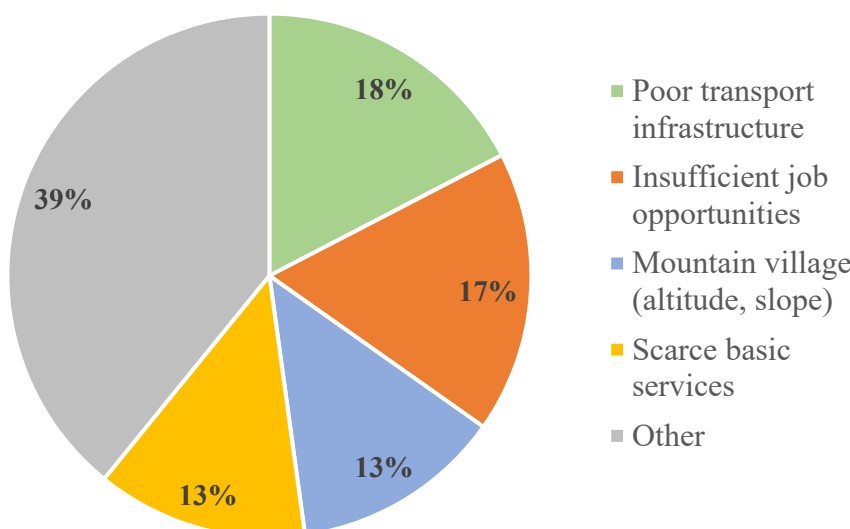

**Figure 4.** Stated causes for the depopulation figures in the two worst performing protected municipalities, according to experts.

3.3.2. Local Populations

All the workshops' attendees from protected municipalities mentioned some causes for the depopulation figures of their municipalities. Among them, the two major ones were 'Insufficient job opportunities' and 'Scarce basic services', accounting for 50% of all the stated causes (Table 3).

**Table 3.** Stated causes for the depopulation figures in protected municipalities, according to local residents.

| Cause | Mentions (%) |
| --- | --- |
| Insufficient job opportunities | 27.93 |
| Scarce basic services (incl. Internet) | 22.52 |
| Transport infrastructure | 8.11 |
| Administrative issues | 6.31 |
| Housing availability | 4.50 |
| Absence of economic incentives | 4.50 |
| Closeness to larger towns or cities | 3.60 |
| Immigration | 3.60 |
| Few births | 2.70 |
| Youngsters remained | 2.70 |
| Scarce land availability | 1.80 |
| Emigration | 1.80 |
| Difference between official population figures and reality | 1.80 |
| Other | 8.11 |

*3.4. Recommendations to Revert Rural Depopulation*

3.4.1. General recommendations

A total of 94.7% of the experts suggested some actions to improve depopulation figures in protected municipalities. The main proposed action by the experts was improving or providing basic services, including high-speed internet (Figure 5).

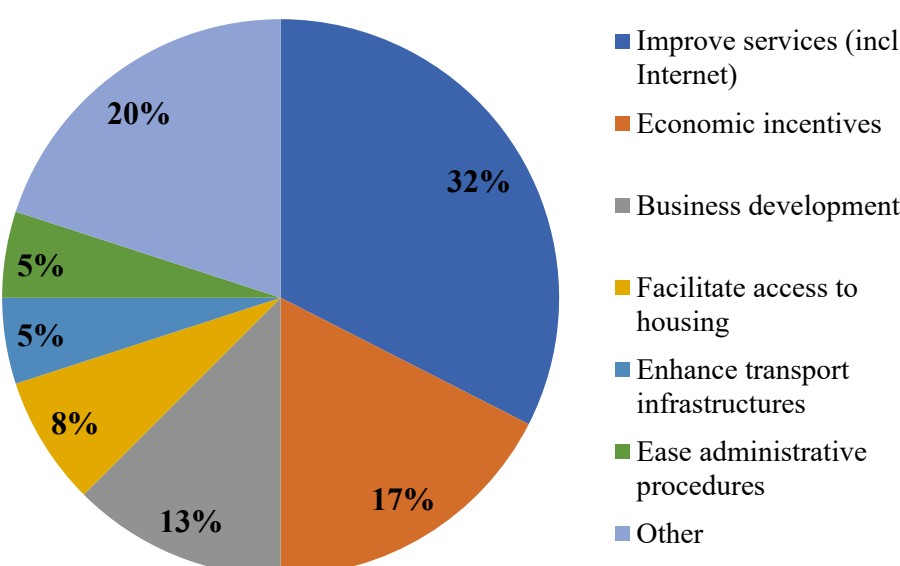

**Figure 5.** Experts' main recommendations to revert depopulation in protected municipalities.

All the attendees to the workshops provided some ideas to improve depopulation figures in their towns. Nearly three quarters of the proposals had to do with providing, improving or maintaining basic services, developing businesses, or creating or providing jobs in general terms (Figure 6).

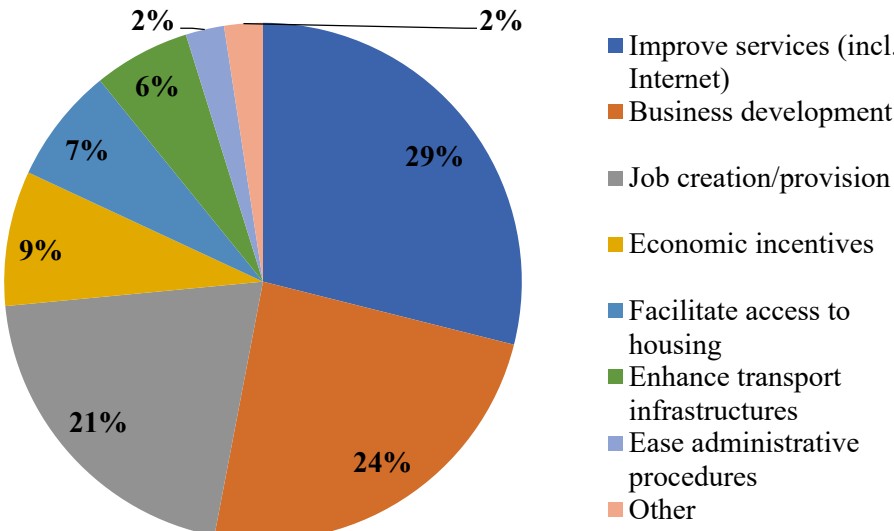

**Figure 6.** Residents' main recommendations to revert depopulation in protected municipalities.

3.4.2. Recommendations regarding Protected Areas

A total of 79% of the experts expressed that PA regulations or management could contribute to improving the municipalities' depopulation figures, chiefly through the greater compatibility of uses, promotion of economic activities and shared management of the sites (Figure 7).

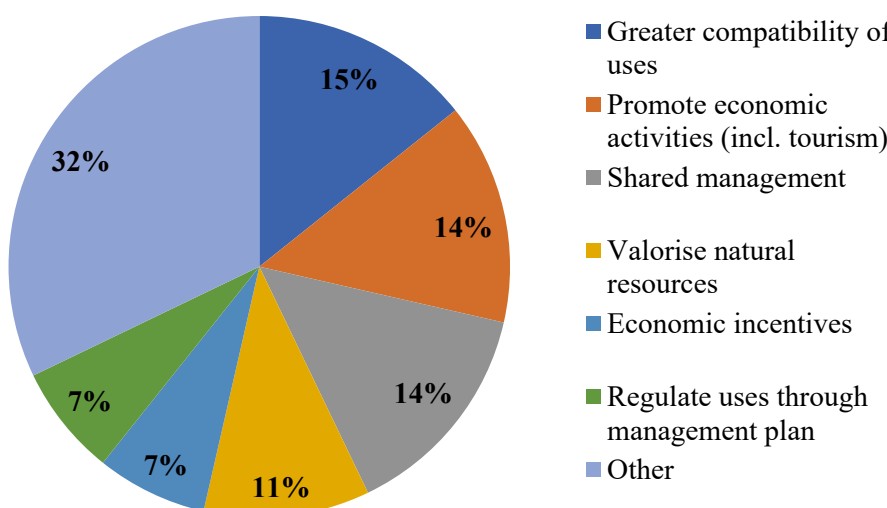

**Figure 7.** Experts' actions to improve depopulation figures in protected municipalities to be implemented by protected areas' administrations.

Half of the workshops' attendees from protected municipalities thought that the regulations or management of their PAs could contribute to improving depopulation figures in their municipalities, mainly through tourism promotion and easing restrictions to development (Figure 8).

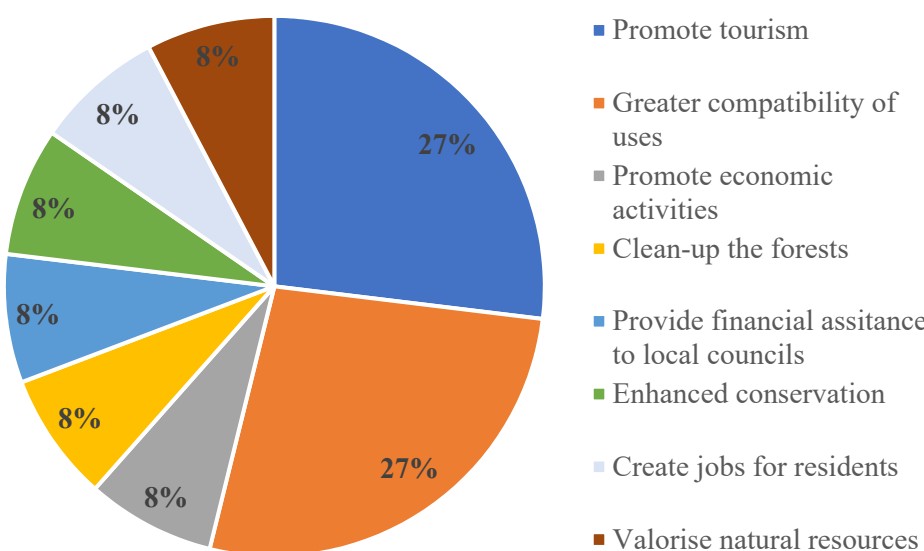

**Figure 8.** Residents' actions to improve depopulation figures in protected municipalities to be implemented by protected areas' administrations.

## 4. Discussion

### 4.1. Effects of PAs on Rural Depopulation

There was broad agreement between residents and experts that PAs had had little or no effect on rural wellbeing and depopulation. Some previous studies had shown little, if any, social or economic impacts at a municipality scale from PA designation on land [22,23] and at sea [24] in European settings. However, for a noticeable number of the workshops' attendees and PA managers, PA designation had had some negative impacts on local wellbeing and depopulation due to restrictions to economic activities. The authors of [25] found similarly negative social perceptions on Natura 2000 sites linked to socioeconomic restrictions and little social engagement in PA designation and management across Europe. These results also align with a previous study using official statistics in Spain where protected municipalities generally performed worse than unprotected ones when facing rural depopulation [17]. Our results, however, contrast with those based on perceptions on European PAs from [18]. In their review, they found mostly positive perceived effects of European PAs on wellbeing and local livelihoods, except for MPAs. In other settings, different authors found varying outcomes of PAs using semi-experimental research designs: [26] stated no demographic or socioeconomic differences between municipalities inside and outside national parks in Colombia, especially after the armed conflict in the country had ended. In turn, [15] found that PAs in Thailand and Costa Rica had improved the economic conditions of local communities, whereas [19] found that communities living closer to a PA in Kenya had experienced greater economic losses than distant communities, although non-statistical differences in welfare or poverty between the two were found.

In contrast to numerous claims [27–29], respondents from both protected and unprotected municipalities in our study did not mostly identify any specific benefits from ecosystem service provision by PAs. Previous studies suggest that the socioeconomic impacts of (M)PAs are mostly felt in rural communities in developing countries with a high dependence on natural resource exploitation [30–32], rather than in more economically diversified settings like Europe. As a result, many developing nations have established compensation schemes for local populations living near PAs to offset restrictions in access to natural resources and economic losses from wildlife [33]. In developed countries, a few specific socioeconomic groups linked to resource extraction, such as fishers, are likely to be the only ones noticeably affected by (M)PA regulations or management [18,34,35]. In our study, very few respondents perceived a positive impact of PAs on local depopulation and wellbeing linked to an increase in tourist activities or ecosystem service provision, as

often suggested for PAs [27,36–38]. Only the most geographically detached participants in our study, including non-PA managers and residents from unprotected municipalities, perceived slightly positive impacts from PA designation overall. This is consistent with previous studies that showed worse perceptions of social and economic effects of PAs by local stakeholders [18,39]. Our sample of respondents were asked about leniently regulated, multiple-use Natura 2000 sites in which a broad range of socioeconomic activities are allowed and even encouraged [40]. Thus, normative restrictions are less likely to be felt in those sites compared to more stringently regulated reserves [41]. Nevertheless, the large numbers of variables that may affect social, economic and demographic variables at a local scale make it challenging to attribute socioeconomic outcomes exclusively to PAs, even if confounders are considered [26,42,43].

No impact of PA management on depopulation was perceived by any of the experts. Actually, the three SCIs were endowed with a management plan recently and became SACs between 2015 and 2016. Such plans include a number of management guidelines and actions to be implemented according to budgetary availability. However, few of those actions have seemingly been implemented yet. In turn, the SPA had no management plan and was not actively managed at the time this study was conducted. A substantial delay in the development of management plans for Natura 2000 sites in Spain meant that a notable number of such sites had not yet been endowed with a management plan by 2020 [44]. Actually, by the end of 2017, the reported overall status of management planning in Natura 2000 sites across Europe differed by country but was overly deficient [45].

Stakeholders' perceptions are considered essential to the assessment and eventual success of conservation initiatives [18,19]. However, they have been found not to always align with objective data [22,39,46]. Therefore, care should be taken when making decisions that are entirely based on perceptions; such decisions should ideally be complemented with reliable statistics.

### 4.2. Other Causes of Rural Depopulation

Three major themes affecting rural depopulation stood out for both experts and residents: transport infrastructure, scarcity of basic services, and insufficient job opportunities, as previously shown for many Eastern and Southern European rural areas [6,47]. The importance of transport networks for rural depopulation has been previously highlighted, with some successful alternatives to private transportation in rural areas being implemented [48]. All the surveyed municipalities (protected or unprotected) showing better depopulation figures after PA designations were connected by national or regional roads, whereas all the municipalities with worse depopulation figures at that time were accessed only by local roads, which complicate access to basic services that are not provided locally [47]. It is noteworthy that the two best-performing protected municipalities had national roads, which were developed before the PAs' designation. It is most likely that improvements in their accessibility would have been more challenging in those sites had the PAs been designated before such roads were made [49,50]. However, linkages between PA regulations and transport infrastructure development did not explicitly arise from the questionnaires.

The scarcity of basic services, including healthcare, primary and secondary education, public transport, food provision and cultural and leisure activities has been a constant claim by Spanish and European rural residents for a long time [6,7,47]. Shortages and little diversity of job offers, mostly linked to the primary sector, have also been a major well-known cause for rural depopulation in many European settings [6]. However, it is worth mentioning that some residents expressed both in the questionnaire and verbally that there were difficulties in finding workers for some physically demanding jobs in their towns, such as bartender or baker. Many of these jobs are covered by immigrant populations [51,52]. Linked to this, short-term, publicly subsidized jobs were thought by some to outcompete privately offered jobs in terms of working hours and salary, thus hampering inner business development. Some residents also mentioned the challenge of seasonality in agriculture-related jobs.

It is remarkable that 'administrative issues' related to excessive bureaucracy, such as long and complex procedures for getting licenses and permits, were the fourth major perceived cause of depopulation in the protected municipalities. It is likely that part of that bureaucracy is related to the presence of the PAs, as administrative procedures tend to be greater in number and harder for PAs [50], especially if overlapping PA categories exist [53]. However, a noticeable number of residents mentioned neglect of developing initiatives and projects by public administrations even inside the towns' urban areas, which also points to governance issues [47].

House availability was mentioned by both experts and residents as a moderate factor for depopulation. This is surprising in places with small or shrinking populations and ample land for development. Little new offers, the poor state of available houses, high prices and a reluctance to sell properties were mentioned as causes limiting house availability in a number of protected municipalities. Of all these causes, PAs are likely influencing new residential offers by restricting new developments, which may in turn contribute to rising prices, although this should happen in contexts of high housing demand, which did not seem to be present in any of the protected municipalities.

### 4.3. Recommendations against Depopulation

Experts and residents generally agreed that the main measures to improve depopulation figures related to basic service provision, business development and economic incentives. Basic service provision through re-opening, restoring or opening new facilities (including remotely provided services) in suitable developed areas should be compatible with PA management and conservation, and relies chiefly on political and economic decisions [47,54,55].

Given the economic inefficiency of having permanent services in every small town, some experts suggested a county-scale scattered provision of basic services so that all rural populations have access to all needed services (e.g. primary and secondary schools, general practitioners, daycare centres, supermarkets, public transport to bigger towns or cities, banking, culture, etc.) within a short distance of around 20km from their places of residence. The overall success of such arrangements is largely dependent on an efficient and frequent public transport system which grants effective access to local services to all residents [56]. A combination of virtual delivery of some services, home delivery of some other services and a diversified territory though a spatially limited network of basic physical services accessed through a sufficiently frequent public transport system seems a feasible compromise between social and economic sustainability for Spanish rural populations [7,47]. Moreover, rural areas should use existing and innovative approaches to valorise their large, varied and unique natural and cultural capitals as a source of endogenous development [47].

Immigration has been shown to reduce or even revert depopulation in some Spanish rural areas [5,57,58], although it is thought not to offset population losses in European rural areas [47]. Young immigrants often take up low-waged, physically demanding, unskilled jobs—some of them outside the formal economy—that many locals are unwilling to do [51], thus helping to maintain population numbers, services, landscapes, and rural traditions [5]. However, attracting foreign workers was not mentioned as a possible solution to depopulation issues either by residents or experts.

The main measures to improve depopulation figures to be taken by the PAs were linked to promoting tourism and greater compatibility between conservation actions and socioeconomic activities. Natura 2000 sites have been devised to be compatible with a broad range of socioeconomic activities [40]. Only impactful activities that may compromise the effective conservation of biodiversity are restricted or forbidden in them [49]. However, some such activities, such as housing or infrastructure development, were demanded by some of our respondents, including some experts, as measures to enhance population figures and rural wellbeing. Natura 2000 sites cover over 17% of the territory of the European Union and include the most valuable places for biodiversity in Europe [9].

Developing those sites can and should only happen exceptionally and should be offset by adequate compensatory measures [49]. Thus, developmental options cannot generally be regarded as feasible solutions for protected rural areas.

In contrast, tourism has been advocated as the great economic benefit from PAs to local communities [36,59,60], although benefit sharing issues are common, especially in developing settings [61]. Moreover, costs to local populations from tourism development such as rising prices or restricted access to land are often overlooked and should also be taken into account [62]. Tourism has actually proved to be beneficial for some protected communities around the world [27,32,37]. Nevertheless, tourism in Natura 2000 sites is still low given the little degree of knowledge of the network by European populations [63,64] and the scarce tourism infrastructure in place in many sites. Even where some visitors' infrastructure is in place, very little data exist on how it is used or how it influences visitors' awareness and behaviour [59]. Moreover, tourism in PAs needs to be carefully planned and managed so that visitors do not jeopardise conservation with their behaviours or numbers [65]. Thus, given that still a notable proportion of Natura 2000 sites in Spain have little or no active management yet [44], broadly promoting tourism in such sensitive areas might be risky from a sustainability viewpoint and should be handled with care.

Social restrictions and health issues linked to the COVID-19 pandemic have caused substantial urban-rural migration in Spain from 2020, leading to population recovery in many rural areas [66]. It is, however, uncertain whether such a trend will remain once the pandemic is controlled and the legal and health situations return to normal.

*4.4. Methodological Remarks*

There are some methodological considerations of this study to be made: firstly, the numerically limited nature of our samples of residents. We managed to survey 6.5% of the official census of residents in the four protected municipalities in 2019. There were, however, some claims by the interviewees that official populations registers are overestimated as some people in the register do not live in town for most of the year. Thus, our sampling percentage would likely be higher.

Factors such as the limited participatory tradition in Spanish rural municipalities, time coincidence with some agricultural works (seeding) and some weather events (some rain and haze happened just before two workshops) are likely to have reduced participation in the workshops [67]. Secondly, the samples of attendees to the workshops were selected with the help of towns' mayors. This is likely to have entailed some bias in the sample of residents towards local-council-related residents and views. The non-random selection of attendees to the workshops meant that the representation of their responses might be compromised and should not be assumed. Nevertheless, we had no other means of reaching local residents other than relying on the town mayors and their dissemination of workshop information.

**5. Conclusions**

The major perceived causes for the protected municipalities' population trends were not related to PA designation or management, but rather to structural causes linked to transport infrastructure, job provision and basic service delivery. Actually, PAs were perceived to play a minor role in terms of rural depopulation and local wellbeing in the selected small rural municipalities.

Some of the proposed measures to improve depopulation figures and rural wellbeing, notably basic service provision and new job opportunities, are generally compatible with biodiversity conservation in PAs, whereas other more 'developmental' measures are not and should be addressed with care, given the sensitive nature of protected biodiversity. Natura 2000 sites could help create new jobs linked to PA management or tourism, but they have little potential for greater compatibility of land uses that respects ecological integrity, given their lenient legal regimes.

Future work would benefit from contrasting stakeholders' perceptions with official statistics in order to obtain a more comprehensive, less biased picture of the depopulation issue in small rural municipalities such as the ones studied here.

**Supplementary Materials:** The following supporting information can be downloaded at: https://www.mdpi.com/article/10.3390/land11030384/s1, Annex S1. Experts' questionnaire, Annex S2. Residents' questionnaire, Annex S3. Guidance to mayors on workshops' participants, Annex S4. Poster announcing the workshop.

**Author Contributions:** Conceptualization, D.R.-R.; methodology, D.R.-R.; formal analysis, D.R.-R.; investigation, D.R.-R.; writing—original draft preparation, D.R.-R.; writing—review and editing, R.L.V.; project administration, R.L.V.; funding acquisition, R.L.V. All authors have read and agreed to the published version of the manuscript.

**Funding:** This study was funded by the University of Malaga through its Research Plan 2020, Research Grant Number B3-2020-04.

**Data Availability Statement:** Not applicable.

**Acknowledgments:** We would like to thank all the respondents to the phone interviews, the eight town mayors, who greatly facilitated the workshops, and the attendees to the workshops for their essential contributions to this work. We would also like to acknowledge two anonymous reviewers whose comments helped to enhance the quality of this text.

**Conflicts of Interest:** The authors declare no conflict of interest. The funders had no role in the design of the study; in the collection, analyses, or interpretation of data; in the writing of the manuscript, or in the decision to publish the results.

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
