# Peer review of "Protected Areas and Rural Depopulation in Spain: A Multi-Stakeholder Perceptual Study"

_land, doi:10.3390/land11030384_

Round 1

Reviewer 1 Report

Dear Authors,

It is an interesting paper. However I suggest to define the research problem and clarify the purpose of the study in the introduction. It would be also interesting if you present in this section the structure of the paper.

In the conclusion section I recommend to relate it with literature review, In the same way add here the theoretical and practical implications of the study as also paths for future research. 

Good work. 

Reviewer 2 Report

It would be interesting if the analysis of the 4 case studies of PA where the inquiry was carried out had also contemplated a more concrete analysis of the realities lived there.

Thus, it would be possible to discern to what extent the topics mentioned by the respondents as less advantageous in the PA are in fact crucial and whether or not these have evolved in the near past (transport system – of goods and people); Internet network provision; limitations to the exercise of productive activities (in which sectors, which ones)…etc.

 It would also be interesting to know to what extent, in the primary sector, access to compensation payments for activities imposed by the Natura 2000 Network through the Rural Development policy is effective and what it represents.

The survey carried out gives us an account of the respondents' perception and their opinion, but questions of fact that allow a more objective assessment of what has actually been operating in the field in these PAs were not evaluated in this information survey, nor is it properly presented through by the authors. In similar field work I have found that respondents' opinions are often biased in relation to what actually happens.
